# Boosting Protein Graph Representations through Static-Dynamic Fusion

**Pengkang Guo**[1]  **Bruno Correia**[1]  **Pierre Vandergheynst**[1]  **Daniel Probst**[1][2]

## Abstract

Machine learning for protein modeling faces significant challenges due to proteins' inherently dynamic nature, yet most graph-based machine learning methods rely solely on static structural information. Recently, the growing availability of molecular dynamics trajectories provides new opportunities for understanding the dynamic behavior of proteins; however, computational methods for utilizing this dynamic information remain limited. We propose a novel graph representation that integrates both static structural information and dynamic correlations from molecular dynamics trajectories, enabling more comprehensive modeling of proteins. By applying relational graph neural networks (RGNNs) to process this heterogeneous representation, we demonstrate significant improvements over structure-based approaches across three distinct tasks: atomic adaptability prediction, binding site detection, and binding affinity prediction. Our results validate that combining static and dynamic information provides complementary signals for understanding protein-ligand interactions, offering new possibilities for drug design and structural biology applications.

## 1. Introduction

With the recent surge and successes of deep learning methods in protein structure prediction, attention is rapidly turning towards the prediction of the temporal behavior of these highly dynamic macromolecules. Combined with quantitative and qualitative advances in molecular dynamics simulations (Joshi & Deshmukh, 2021; Zeng et al., 2021; Majewski et al., 2023; Nam & Wolf-Watz, 2023), this attention is resulting in the increased availability and accessibility of simulated molecular dynamics trajectories (Vander Meer-

sche et al., 2024; Siebenmorgen et al., 2024a; Liu et al., 2024). Consequently, various approaches are being developed to train predictive and generative models capable of producing molecular dynamics trajectories or sample specific conformations (López-Correa et al., 2023; Jing et al., 2024; Lewis et al., 2024). So far, the potential of these increasingly large trajectory datasets to enhance property predictions in proteins and protein-ligand complexes, such as binding site identification and affinity prediction, remains largely unexplored (Dhakal et al., 2022).

Despite these advances, representing and exploiting molecular dynamics trajectories of proteins for machine learning remains challenging due to the diverse and complex nature of protein structures. One effective alternative is to focus on a higher-order representation of protein dynamics through correlation patterns derived from molecular motion. These dynamic correlations are essential to protein function, and the resulting correlation matrices have long been used to analyze protein dynamics (Agarwal et al., 2002; Lange & Grubmüller, 2008).

In this work, we propose combining molecular structure and simulated molecular trajectories through residue-based correlation matrices and relational graph neural networks (Schlichtkrull et al., 2017). We show that this approach enables the exploitation of the rapidly expanding collection of readily available protein dynamics trajectories for protein and protein–ligand property prediction. In summary:

- We propose a novel heterogeneous graph representation that integrates both static structural information and dynamic correlations from molecular trajectories, enabling more comprehensive modeling of protein properties.

- We introduce the first application of relational graph neural networks to directly process molecular dynamics trajectories, demonstrating clear benefits over graph neural networks (GNNs) based on structure alone.

- We validate our approach across three distinct tasks: atomic adaptability prediction, binding site detection, and binding affinity prediction, showing consistent benefits of combining static and dynamic information.

[1]École Polytechnique Fédérale de Lausanne, Lausanne, Switzerland [2]Wageningen University & research, Wageningen, The Netherlands. Correspondence to: Pengkang Guo <pengkang.guo@epfl.ch>.

*Proceedings of the 42$^{nd}$ International Conference on Machine Learning*, Vancouver, Canada. PMLR 267, 2025. Copyright 2025 by the author(s).

## 2. Related Work

### 2.1. Dynamic Correlations in Protein Analysis

Dynamic correlations can be derived through various approaches, including methods like the Gaussian Network Model that use coarse-grained representations and harmonic approximation (Bahar et al., 1997), as well as from molecular dynamics trajectories. The latter approach has been extensively applied in protein analysis, particularly for understanding allosteric mechanisms and signal propagation (Mc-Clendon et al., 2009; Long & Brüschweiler, 2011; Wang et al., 2020), investigating tRNA-protein complex interactions (Sethi et al., 2009), and identifying catalytically important regions for enzyme engineering (Bunzel et al., 2021; Gao et al., 2024).

However, they have not been used as a representation of trajectories when training predictive models on large data sets but mainly as a means to investigate the propagation of structural changes in a single, or a class of proteins through methods such as dynamical network analysis (Melo et al., 2020; Calvó-Tusell et al., 2022).

### 2.2. Machine Learning for Protein Structure and Dynamics

Machine learning for protein structure and dynamics has primarily focused on graph-based methods, though other paradigms are also emerging. Graph neural networks have been widely applied to predict properties and functions of proteins as well as properties of protein-ligand or protein-protein interactions based on structure (Gligorijević et al., 2021a; Li et al., 2021b; Réau et al., 2023). More recently, they have been used to enhance and accelerate molecular dynamics simulations (Wang et al., 2022; Yue et al., 2024).

Chiang et al. (2022) explored incorporating dynamic information into protein graphs by using normal mode analysis to generate correlation edges, combining this with 1D and 2D persistence diagrams of $\alpha$-carbons for molecular function classification using graph convolutional networks (GCN) (Defferrard et al., 2017; Kipf & Welling, 2017).

Relational graph neural networks have also shown promise in small molecular graph generation (Zou et al., 2023), and protein representation learning, integrating sequential and spatial distance in proteins (Zhang et al., 2022).

Beyond graph-based approaches, Sun et al. (2023) introduced Dynamical Surface Representation, which uses implicit neural networks to model protein dynamics through continuous surface representations, enabling scalable modeling of large protein conformational changes.

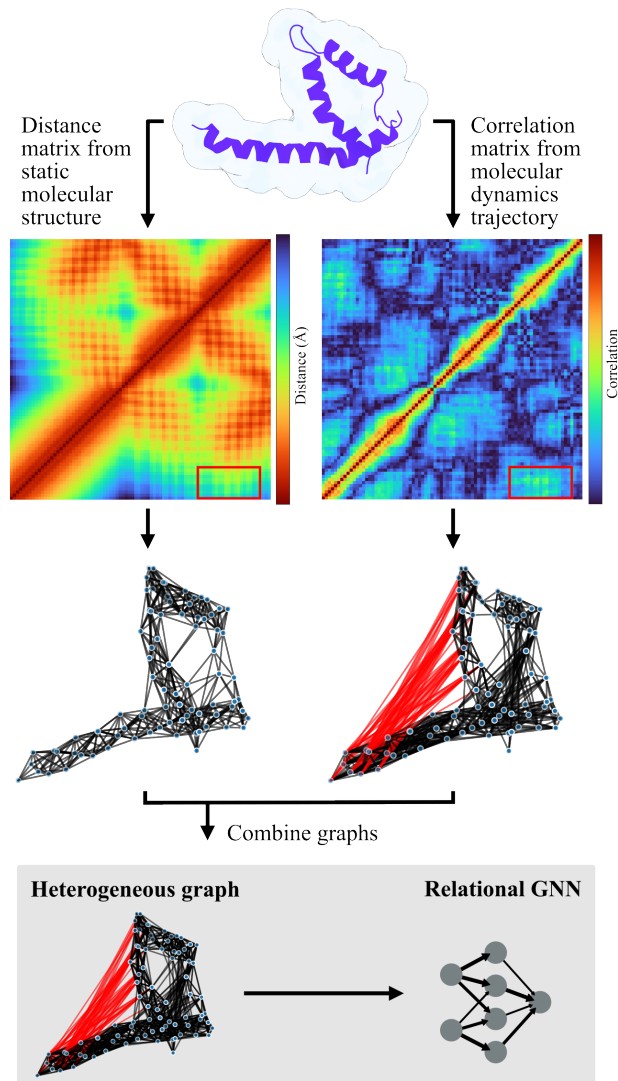

*Figure 1.* **Fusion of static structure and molecular dynamics information**. The left side shows the transformation of protein structure (PDB ID 5GMU) into a distance-based matrix, while the right side presents the correlation matrix derived from molecular dynamics trajectories, which shows motion correlations between different regions. The correlation edges create direct connections between dynamically coupled regions that may be spatially distant (shown in red), enabling efficient information flow across the protein structure. The fusion of these structural and dynamic features creates a heterogeneous graph representation, which serves as input to relational graph neural networks.

# 3. Methodology

## 3.1. Graph Construction Framework

We represent a protein complex as a tuple $(V, E_d, E_c)$, where $V$ represents the set of nodes, $E_d$ represents distance-based edges, and $E_c$ represents correlation-based edges derived from molecular dynamics trajectories.

We present a novel approach for incorporating both static structural and dynamic motion information into protein graph representations. Our method consists of two key components: (1) a heterogeneous graph construction framework that combines spatial proximity with dynamic correlations from molecular dynamics simulations, and (2) the application of relational neural networks to effectively process the heterogeneous graphs enriched by both structural and dynamic information.

As illustrated in Figure 1, our approach derives two complementary edge types from protein data: distance edges based on static structure, and correlation edges from molecular dynamics trajectories. These correlation edges provide direct links between dynamically coupled regions of the protein, enabling more efficient long-range information flow than in the original graph structure. The mechanism is akin to graph rewiring, which is known to mitigate over-squashing in GNNs (Topping et al., 2021) (see Appendix A.6 for further analysis).

### 3.1.1. NODE DEFINITION AND FEATURES

Nodes are defined based on the specific requirements of each task:

For atomic-level predictions (e.g., atomic property prediction), each node represents a non-hydrogen atom, capturing detailed molecular interactions at the atomic scale.

For residue-level tasks (e.g., binding site detection), each node represents a residue, where the coordinates of its $C_\alpha$ atom are used to determine the residue's spatial position.

Each node $v_i \in V$ is associated with a feature vector $\mathbf{h}_i \in \mathbb{R}^d$ consisting of the one-hot encoding of the atom/residue type and the atom charge (for atomic-level graphs).

### 3.1.2. DISTANCE-BASED EDGE CONSTRUCTION

The distance-based edges $E_d$ capture spatial proximity in the static structure:

$$E_d = (v_i, v_j) \mid d(v_i, v_j) < \tau_d \tag{1}$$

where $d(v_i, v_j)$ represents the Euclidean distance between nodes, and $\tau_d$ is a distance threshold (4.5 Å for atomic-level and 10 Å for residue-level graphs). These thresholds are widely used in protein modeling: the 4.5 Å threshold cap-

tures meaningful atomic interactions (Bouysset & Fiorucci, 2021), while the 10 Å threshold is commonly adopted for residue-level contacts (Gligorijević et al., 2021b).

### 3.1.3. DYNAMIC CORRELATION EDGE CONSTRUCTION

To capture dynamic behaviors, we analyze molecular dynamics trajectories to construct correlation-based edges $E_c$. Before computing correlations, all trajectory frames are aligned to the initial structure through rigid-body superposition optimized to minimize the root-mean-square deviation (RMSD) between equivalent atomic positions. The alignment eliminates global translations and rotations while preserving internal conformational changes.

Unlike distance-based representations that primarily capture local structural relationships, correlation-based edges can identify dynamically coupled regions regardless of spatial proximity, creating direct pathways between motion-related but spatially distant parts of the protein (as shown in Figure 1). For each pair of nodes, we compute their motion correlation across simulation frames:

$$C_{ij} = \frac{1}{T} \sum_{t=1}^{T} \frac{\Delta \mathbf{r}_i^t \cdot \Delta \mathbf{r}_j^t}{|\Delta \mathbf{r}_i^t||\Delta \mathbf{r}_j^t|} \tag{2}$$

where $\Delta \mathbf{r}_i^t$ represents the displacement vector of node $i$ at frame $t$, and $T$ is the total number of frames. The correlation edges are then defined as:

$$E_c = (v_i, v_j) \mid |C_{ij}| > \tau_c \tag{3}$$

where $\tau_c$ is the correlation threshold (0.6 for atomic-level and 0.3 for residue-level graphs). These thresholds are chosen to maintain similar graph sparsity, thereby achieving a fairer comparison when either Correlation or Distance Graph is used.

### 3.1.4. COMBINED GRAPH

The final graph representation integrates both distance-based and correlation-based edges, yielding a heterogeneous graph that captures both static structural information and dynamic behavioral patterns. This combined representation enables the model to utilize local spatial relationships and potential long-range dynamic interactions simultaneously.

## 3.2. Relational Graph Neural Network Architecture

The heterogeneous nature of our Combined Graph, containing both distance-based and correlation-based edges, requires a neural network architecture capable of processing different types of relationships. We therefore employ two established relational neural networks: the Relational Graph Convolutional Network (RGCN) (Schlichtkrull et al.,

2018) and the Relational Graph Attention Network (RGAT) (Busbridge et al., 2019). These architectures are particularly suited for our approach as they handle heterogeneous edges by learning different weight matrices for different edge types.

The RGCN extends the traditional Graph Convolutional Network by introducing relation-specific transformations. For each layer $l$, the message passing operation is defined as:

$$\mathbf{h}_i^{(l+1)} = \sigma(\sum_{r \in \mathcal{R}} \sum_{j \in \mathcal{N}_r(i)} \frac{1}{|\mathcal{N}_r(i)|} \mathbf{W}_r^{(l)} \mathbf{h}_j^{(l)} + \mathbf{W}_0^{(l)} \mathbf{h}_i^{(l)})$$

(4)

where $\mathcal{N}_r(i)$ denotes neighbors of node $i$ connected by edges of type $r$, $\mathbf{W}_r^{(l)}$ is the relation-specific transformation matrix, and $\mathbf{W}_0^{(l)}$ is the self-connection weight matrix. In our case, $\mathcal{R}$ represents the set of edge types (distance and correlation). This formulation allows the network to learn distinct transformations for distance-based and correlation-based relationships, enabling it to capture the unique characteristics of each edge type.

The RGAT extends this formulation by incorporating an attention mechanism. This formulation allows the network to learn distinct transformations for distance-based and correlation-based relationships, enabling it to capture the unique characteristics of each edge type. For each layer $l$, the attention-based message passing operation is defined as:

$$\mathbf{h}_i^{(l+1)} = \sigma(\sum_{r \in \mathcal{R}} \sum_{j \in \mathcal{N}_r(i)} \alpha_{ij}^{(r)} \mathbf{W}_r^{(l)} \mathbf{h}_j^{(l)} + \mathbf{W}_0^{(l)} \mathbf{h}_i^{(l)}) \quad (5)$$

The attention coefficients $\alpha_{ij}^{(r)}$ are computed using query and key kernels for each relation type $r$:

$$\mathbf{q}_i^{(r)} = \mathbf{W}_1^{(r)} \mathbf{x}_i \cdot \mathbf{Q}^{(r)} \quad \text{and} \quad \mathbf{k}_i^{(r)} = \mathbf{W}_1^{(r)} \mathbf{x}_i \cdot \mathbf{K}^{(r)} \quad (6)$$

These kernels are used to compute attention logits:

$$\mathbf{a}_{i,j}^{(r)} = \text{LeakyReLU}(\mathbf{q}_i^{(r)} + \mathbf{k}_j^{(r)}) \quad (7)$$

The final attention coefficients are obtained as:

$$\alpha_{i,j}^{(r)} = \frac{\exp(\mathbf{a}_{i,j}^{(r)})}{\sum_{r' \in \mathcal{R}} \sum_{k \in \mathcal{N}_{r'}(i)} \exp(\mathbf{a}_{i,k}^{(r')})} \quad (8)$$

This attention mechanism enables the model to automatically determine the relative importance of different relationships, potentially providing insights into the contributions of structural and dynamic information in protein modeling.

To validate the generalizability of our approach, we also evaluate three additional architectures representing different paradigms: EGNN (Satorras et al., 2021) as a representative equivariant GNN, GPS (Rampášek et al., 2022) as a popular graph transformer, and SS-GNN (Zhang et al., 2023) as a domain-specific model for binding affinity prediction. We create relational variants (R-EGNN, R-GPS, and R-SS-GNN, respectively) as follows: for EGNN, we process distance and correlation graphs with separate models and merge their outputs; for GPS, we replace its local message passing layer with RGCN; for SS-GNN, we similarly replace the GNN component with RGCN while maintaining the original hyperparameters and featurization.

# 4. Experiments

## 4.1. Dataset

We evaluate our approach using the MISATO dataset (Siebenmorgen et al., 2024b), which contains 19,443 protein-ligand complexes derived from PDBbind (Su et al., 2018; Liu et al., 2017; Wang et al., 2005). Each complex undergoes semi-empirical quantum mechanical refinement and 10 ns molecular dynamics simulation using the Amber20 software package. The dataset also provides key physico-chemical properties, forming a high-quality benchmark for machine learning tasks.

To ensure robust evaluation and prevent information leakage through structural similarities, the dataset is split into training (80%), validation (10%), and test (10%) sets using protein sequence clustering via BlastP, ensuring that proteins with high sequence similarity are assigned to the same split (details in Appendix A.3).

## 4.2. Experimental Setup

As discussed in Section 3.2, we use RGCN, RGAT, R-EGNN, R-GPS, and R-SS-GNN to validate our approach. We evaluate three tasks: (1) atom adaptability prediction, (2) binding site detection, and (3) binding affinity prediction. For each task, we evaluate three graph representations: Distance Graph based on the static structure, Correlation Graph derived from the molecular dynamics trajectory, and Combined Graph that integrates both sources. To ensure a fair comparison, we maintain identical input node features and model architectures across all graph types for all tasks, with edge definitions being the only variable, allowing us to validate the role of dynamic information.

## 4.3. Results and Discussion

We evaluate our approach on three distinct prediction tasks: atomic adaptability, binding site identification, and binding affinity prediction. For each task, we analyze how differ-

ent graph representations (Distance, Correlation, and Combined) affect model performance using the various architectures.

### 4.3.1. ATOMIC ADAPTABILITY PREDICTION

Atomic adaptability quantifies the conformational plasticity of atoms within a protein structure, where higher values indicate greater flexibility and lower values indicate rigidity (see Appendix A.4). This property helps identify key regions of motion, making it crucial for understanding protein dynamics and molecular design. We formulate adaptability prediction as a node-level regression task, where each atom is annotated with an adaptability score from the MISATO dataset (see Figure 2).

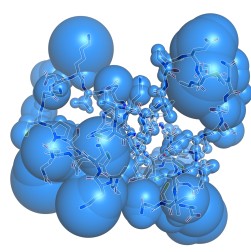 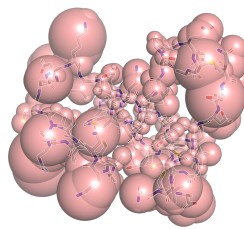

*Figure 2.* **Atomic Adaptability Prediction.** Visualization of per-atom adaptability in a protein structure (PDB ID `5C11`). Left: ground-truth (target) adaptability values shown as blue spheres. Right: predicted adaptability values shown as pink spheres. Sphere size indicates the magnitude of adaptability, with larger spheres corresponding to more flexible (higher adaptability) regions.

Table 1 presents the performance comparison across different graph representations using four architectures. The Correlation Graph, which captures dynamic motion patterns derived from molecular dynamicss trajectories, consistently outperforms the Distance Graph across all metrics and architectures. Using RGCN, we observe improvements in both error metrics (MAE reduces from 0.2658 to 0.2311) and correlation coefficients (Pearson R increases from 0.5259 to 0.6426). Similar comprehensive improvements are observed with RGAT, R-EGNN, and R-GPS, suggesting the value of dynamic information for atomic adaptability prediction.

When both types of information are integrated in the Combined Graph, we observe further significant improvements across all metrics: the Pearson correlation coefficient reaches 0.7326 (RGCN) and 0.7153 (RGAT), representing improvements of 39.3% and 50.7% respectively over the Distance Graph baseline (0.5259 using RGCN and 0.4746 using RGAT). Similar improvements are observed across other metrics, as evidenced by the reduction in MAE from 0.2658 to 0.1981 (RGCN) and 0.2766 to 0.2074 (RGAT). R-GPS follows the same pattern, with the Combined Graph

achieving strong performance across all metrics. Notably, R-EGNN shows a different pattern where the Correlation Graph alone achieves the best performance, while the Combined Graph performs similarly to the Distance Graph. We attribute this to limitations in our preliminary R-EGNN implementation, which may not optimally fuse information from different relation types. We leave the exploration of improved fusion mechanisms for equivariant architectures to future work.

These performance improvements align with the physical nature of atomic adaptability. While spatial proximity (captured by the Distance Graph) provides important structural constraints, atomic adaptability is inherently a dynamical property that highly depends on atomic fluctuations and conformational changes, which cannot be fully captured by spatial proximity alone. The Correlation Graph, leveraging dynamical information derived from molecular dynamics trajectories, better captures elements tied to motion and complements the structural information. When combined, these two edge types enable the model to learn from both spatial constraints and dynamic coupling patterns, resulting in the Combined Graph's superior performance.

The consistent improvement across multiple architectures suggests that the performance gains primarily stem from the richer graph structure rather than specific architectural choices. This robustness validates our approach of incorporating dynamic information through correlation edges as an effective strategy for enhancing protein graph representations in dynamical property prediction.

### 4.3.2. BINDING SITE DETECTION

Binding site detection aims to identify key residues in proteins that directly interact with ligands, specifically those residues within 10 Å from the ligand, following PDBbind. This task is essential for understanding protein functionality and facilitating early-stage drug design. We formulate this as a binary node classification problem at the residue level, where each node represents a residue and is classified as either a binding site or a non-binding site (see Figure 3).

Table 2 presents the classification performance across different graph representations using four architectures. The Combined Graph consistently achieves the best performance across all metrics for all architectures. For RGCN, the F1 score increases from 0.2428 (Distance Graph) and 0.2578 (Correlation Graph) to 0.2834, representing improvements of 16.7% and 9.9% respectively. Similar patterns emerge with RGAT, where the F1 score improves from 0.2089 (Distance Graph) and 0.2294 (Correlation Graph) to 0.2574. R-EGNN and R-GPS show the same pattern of Combined Graph achieving the best results, with R-EGNN achieving notably higher performance, reaching an F1 score of 0.4142 with the Combined Graph.

*Table 1.* **Atomic Adaptability Prediction**. Node-level regression task predicting atomic adaptability values using data from the MISATO dataset. Results show mean $\pm$ standard deviation over 5 runs ($\downarrow$ indicates lower is better, $\uparrow$ indicates higher is better). Notably, the Correlation Graph alone achieves better performance than the Distance Graph on all metrics, suggesting the importance of dynamic information for this task.

| MODEL | GRAPH TYPE | MAE ($\downarrow$) | RMSE ($\downarrow$) | PEARSON R ($\uparrow$) | SPEARMAN R ($\uparrow$) |
|---|---|---|---|---|---|
| RGCN | DISTANCE | $0.2658 \pm 0.0061$ | $0.4274 \pm 0.0008$ | $0.5259 \pm 0.0015$ | $0.5543 \pm 0.0017$ |
| | CORRELATION | $0.2311 \pm 0.0014$ | $0.3846 \pm 0.0011$ | $0.6426 \pm 0.0026$ | $0.6990 \pm 0.0019$ |
| | COMBINED | $\mathbf{0.1981} \pm 0.0020$ | $\mathbf{0.3417} \pm 0.0008$ | $\mathbf{0.7326} \pm 0.0014$ | $\mathbf{0.7922} \pm 0.0010$ |
| RGAT | DISTANCE | $0.2766 \pm 0.0038$ | $0.4419 \pm 0.0018$ | $0.4746 \pm 0.0066$ | $0.4762 \pm 0.0085$ |
| | CORRELATION | $0.2443 \pm 0.0013$ | $0.3976 \pm 0.0013$ | $0.6106 \pm 0.0037$ | $0.6521 \pm 0.0068$ |
| | COMBINED | $\mathbf{0.2074} \pm 0.0030$ | $\mathbf{0.3511} \pm 0.0018$ | $\mathbf{0.7153} \pm 0.0033$ | $\mathbf{0.7699} \pm 0.0024$ |
| R-EGNN | DISTANCE | $0.2305 \pm 0.0047$ | $0.3807 \pm 0.0007$ | $0.6530 \pm 0.0012$ | $0.6914 \pm 0.0015$ |
| | CORRELATION | $\mathbf{0.2028} \pm \mathbf{0.0055}$ | $\mathbf{0.3387} \pm 0.0006$ | $\mathbf{0.7398} \pm 0.0016$ | $\mathbf{0.7710} \pm 0.0033$ |
| | COMBINED | $0.2321 \pm 0.0053$ | $0.3809 \pm 0.0016$ | $0.6532 \pm 0.0018$ | $0.6917 \pm 0.0012$ |
| R-GPS | DISTANCE | $0.2552 \pm 0.0098$ | $0.4232 \pm 0.0015$ | $0.5420 \pm 0.0091$ | $0.5704 \pm 0.0136$ |
| | CORRELATION | $0.2293 \pm 0.0028$ | $0.3806 \pm 0.0009$ | $0.6543 \pm 0.0020$ | $0.7066 \pm 0.0017$ |
| | COMBINED | $\mathbf{0.1921} \pm 0.0009$ | $\mathbf{0.3434} \pm 0.0017$ | $\mathbf{0.7361} \pm 0.0018$ | $\mathbf{0.7961} \pm 0.0013$ |

*Table 2.* **Binding Site Detection**. Node-level binary classification task identifying binding site residues (those within 10 Å from the ligand) using data from the MISATO dataset. Results show mean $\pm$ standard deviation over 5 runs ($\uparrow$ indicates higher is better). The Combined Graph demonstrates superior performance across all metrics and architectures.

| MODEL | GRAPH TYPE | ACC ($\uparrow$) | PRECISION ($\uparrow$) | RECALL ($\uparrow$) | F1 SCORE ($\uparrow$) |
|---|---|---|---|---|---|
| RGCN | DISTANCE | $0.7112 \pm 0.0092$ | $0.1678 \pm 0.0024$ | $0.4464 \pm 0.0164$ | $0.2428 \pm 0.0027$ |
| | CORRELATION | $0.7282 \pm 0.0069$ | $0.1808 \pm 0.0022$ | $0.4552 \pm 0.0102$ | $0.2578 \pm 0.0012$ |
| | COMBINED | $\mathbf{0.7433} \pm 0.0067$ | $\mathbf{0.2005} \pm 0.0030$ | $\mathbf{0.4889} \pm 0.0111$ | $\mathbf{0.2834} \pm 0.0023$ |
| RGAT | DISTANCE | $0.6602 \pm 0.0120$ | $0.1475 \pm 0.0032$ | $0.4439 \pm 0.0234$ | $0.2089 \pm 0.0040$ |
| | CORRELATION | $0.6938 \pm 0.0111$ | $0.1664 \pm 0.0031$ | $0.4441 \pm 0.0182$ | $0.2294 \pm 0.0030$ |
| | COMBINED | $\mathbf{0.7226} \pm 0.0067$ | $\mathbf{0.1861} \pm 0.0029$ | $\mathbf{0.4750} \pm 0.0137$ | $\mathbf{0.2574} \pm 0.0032$ |
| R-EGNN | DISTANCE | $0.8038 \pm 0.0131$ | $0.2749 \pm 0.0110$ | $0.5368 \pm 0.0359$ | $0.3617 \pm 0.0049$ |
| | CORRELATION | $0.7628 \pm 0.0136$ | $0.2181 \pm 0.0043$ | $0.4929 \pm 0.0356$ | $0.3012 \pm 0.0037$ |
| | COMBINED | $\mathbf{0.8387} \pm 0.0149$ | $\mathbf{0.3353} \pm 0.0191$ | $\mathbf{0.5498} \pm 0.0414$ | $\mathbf{0.4142} \pm 0.0043$ |
| R-GPS | DISTANCE | $0.7574 \pm 0.0055$ | $0.2065 \pm 0.0023$ | $0.4675 \pm 0.0131$ | $0.2856 \pm 0.0023$ |
| | CORRELATION | $0.7480 \pm 0.0051$ | $0.2051 \pm 0.0021$ | $0.4935 \pm 0.0093$ | $0.2890 \pm 0.0008$ |
| | COMBINED | $\mathbf{0.7567} \pm 0.0060$ | $\mathbf{0.2274} \pm 0.0029$ | $\mathbf{0.5594} \pm 0.0142$ | $\mathbf{0.3228} \pm 0.0022$ |

Compared to atomic adaptability, which relies more directly on dynamic information, binding site identification depends heavily on static structural features such as protein surfaces and binding pockets. The varying performance patterns between Distance and Correlation graphs across different architectures reflect the balanced importance of both static and dynamic features in this context, with some architectures better suited for leveraging specific types of information. When combined, the model can utilize both spatial proximity and motion patterns, leading to more accurate binding site identification.

The consistent improvement across various architectures demonstrates that these improvements result from the complementary nature of static and dynamic features rather than specific architectural choices. These results validate our

approach of incorporating both types of information into protein graph representations, offering new possibilities for studying complex protein-ligand interactions.

### 4.3.3. BINDING AFFINITY PREDICTION

Binding affinity prediction represents a critical task in drug design and virtual screening, as it quantifies the interaction strength between proteins and ligands. We formulate this as a graph-level regression task, where each graph represents a protein pocket and its ligand, with experimentally measured binding affinities as targets. Following previous work (Li et al., 2021a), we evaluate our approach on the PDBbind 2020 benchmark (details in Appendix A.3).

Table 3 presents the regression performance across different graph representations using five architectures. The Com-

*Table 3.* **Binding Affinity Prediction**. Graph-level regression task predicting protein-ligand binding affinity values using MISATO and PDBbind datasets. Results show mean $\pm$ standard deviation over 5 runs ($\downarrow$ indicates lower is better, $\uparrow$ indicates higher is better). The Combined Graph consistently yields improvements across all metrics and architectures, demonstrating the complementary value of static and dynamic information for binding affinity prediction.

| MODEL | GRAPH TYPE | MAE ($\downarrow$) | RMSE ($\downarrow$) | PEARSON R ($\uparrow$) | SPEARMAN R ($\uparrow$) |
|---|---|---|---|---|---|
| RGCN | DISTANCE | $1.3046 \pm 0.0267$ | $1.6653 \pm 0.0336$ | $0.6596 \pm 0.0156$ | $0.6352 \pm 0.0234$ |
| | CORRELATION | $1.3572 \pm 0.0792$ | $1.6974 \pm 0.0827$ | $0.6360 \pm 0.0428$ | $0.6185 \pm 0.0440$ |
| | COMBINED | $\mathbf{1.2439} \pm 0.0256$ | $\mathbf{1.5798} \pm 0.0447$ | $\mathbf{0.6983} \pm 0.0193$ | $\mathbf{0.6773} \pm 0.0208$ |
| RGAT | DISTANCE | $1.3028 \pm 0.0261$ | $1.6427 \pm 0.0459$ | $0.6694 \pm 0.0222$ | $0.6417 \pm 0.0225$ |
| | CORRELATION | $1.3249 \pm 0.0341$ | $1.6623 \pm 0.0335$ | $0.6643 \pm 0.0212$ | $0.6481 \pm 0.0254$ |
| | COMBINED | $\mathbf{1.2596} \pm 0.0290$ | $\mathbf{1.6012} \pm 0.0411$ | $\mathbf{0.6931} \pm 0.0157$ | $\mathbf{0.6752} \pm 0.0157$ |
| R-EGNN | DISTANCE | $1.2900 \pm 0.0484$ | $1.6614 \pm 0.0635$ | $0.6721 \pm 0.0265$ | $0.6502 \pm 0.0265$ |
| | CORRELATION | $1.4097 \pm 0.0334$ | $1.7888 \pm 0.0585$ | $0.5987 \pm 0.0248$ | $0.5720 \pm 0.0199$ |
| | COMBINED | $\mathbf{1.2632} \pm 0.0463$ | $\mathbf{1.6176} \pm 0.0394$ | $\mathbf{0.6876} \pm 0.0178$ | $\mathbf{0.6773} \pm 0.0206$ |
| R-GPS | DISTANCE | $1.2444 \pm 0.0183$ | $1.5832 \pm 0.0336$ | $0.7026 \pm 0.0104$ | $0.6814 \pm 0.0130$ |
| | CORRELATION | $1.2800 \pm 0.0418$ | $1.6216 \pm 0.0561$ | $0.6862 \pm 0.0155$ | $0.6735 \pm 0.0168$ |
| | COMBINED | $\mathbf{1.2127} \pm 0.0365$ | $\mathbf{1.5326} \pm 0.0521$ | $\mathbf{0.7197} \pm 0.0212$ | $\mathbf{0.7080} \pm 0.0266$ |
| R-SS-GNN | DISTANCE | $1.2378 \pm 0.0378$ | $1.4306 \pm 0.0491$ | $0.7638 \pm 0.0205$ | $0.7454 \pm 0.0256$ |
| | CORRELATION | $1.2278 \pm 0.0330$ | $1.5345 \pm 0.0397$ | $0.7229 \pm 0.0181$ | $0.7015 \pm 0.0254$ |
| | COMBINED | $\mathbf{1.0874} \pm 0.0430$ | $\mathbf{1.3658} \pm 0.0504$ | $\mathbf{0.7873} \pm 0.0187$ | $\mathbf{0.7792} \pm 0.0221$ |

*Table 4.* **Effect of Trajectory Alignment on Atomic Adaptability Prediction**. Comparison of atomic adaptability prediction results using aligned versus unaligned molecular dynamics trajectories. Results show mean $\pm$ standard deviation over 5 runs ($\downarrow$ indicates lower is better, $\uparrow$ indicates higher is better). Trajectory alignment, performed by minimizing RMSD between frames, consistently enhances performance across all metrics for both Correlation and Combined graphs, highlighting the importance of isolating intrinsic conformational dynamics from global rigid-body motions.

| MODEL | GRAPH TYPE | MAE ($\downarrow$) | RMSE ($\downarrow$) | PEARSON R ($\uparrow$) | SPEARMAN R ($\uparrow$) |
|---|---|---|---|---|---|
| RGCN | UNALIGNED CORRELATION | $0.2591 \pm 0.0050$ | $0.4198 \pm 0.0010$ | $0.5493 \pm 0.0024$ | $0.5682 \pm 0.0014$ |
| | ALIGNED CORRELATION | $\mathbf{0.2311} \pm 0.0014$ | $\mathbf{0.3846} \pm 0.0011$ | $\mathbf{0.6426} \pm 0.0026$ | $\mathbf{0.6990} \pm 0.0019$ |
| | UNALIGNED COMBINED | $0.2192 \pm 0.0030$ | $0.3703 \pm 0.0017$ | $0.6762 \pm 0.0026$ | $0.7016 \pm 0.0032$ |
| | ALIGNED COMBINED | $\mathbf{0.1981} \pm 0.0020$ | $\mathbf{0.3417} \pm 0.0008$ | $\mathbf{0.7326} \pm 0.0014$ | $\mathbf{0.7922} \pm 0.0010$ |
| RGAT | UNALIGNED CORRELATION | $0.2653 \pm 0.0024$ | $0.4350 \pm 0.0020$ | $0.5001 \pm 0.0076$ | $0.5194 \pm 0.0073$ |
| | ALIGNED CORRELATION | $\mathbf{0.2443} \pm 0.0013$ | $\mathbf{0.3976} \pm 0.0013$ | $\mathbf{0.6106} \pm 0.0037$ | $\mathbf{0.6521} \pm 0.0068$ |
| | UNALIGNED COMBINED | $0.2274 \pm 0.0017$ | $0.3759 \pm 0.0018$ | $0.6633 \pm 0.0037$ | $0.6914 \pm 0.0037$ |
| | ALIGNED COMBINED | $\mathbf{0.2074} \pm 0.0030$ | $\mathbf{0.3511} \pm 0.0018$ | $\mathbf{0.7153} \pm 0.0033$ | $\mathbf{0.7699} \pm 0.0024$ |

bined Graph consistently achieves the best performance across all metrics and architectures, demonstrating the value of fusing both types of information. Using RGCN, the Combined Graph reaches a Pearson correlation of 0.6983 and reduces MAE to 1.2439, improving upon both the Distance Graph (0.6596, 1.3046) and Correlation Graph (0.6360, 1.3572). Similar patterns emerge with RGAT, where the Combined Graph achieves a Pearson correlation of 0.6931 and MAE of 1.2596, outperforming both single-information approaches (Distance Graph: 0.6694, 1.3028; Correlation Graph: 0.6643, 1.3249). R-EGNN, R-GPS, and R-SS-GNN show the same pattern of Combined Graph achieving the best results.

These results reflect the complex nature of protein-ligand binding affinity, which requires both structural and dynamic information for accurate prediction. While static distance information captures essential geometric constraints, it cannot reflect potential conformational adjustments and long-range interactions during binding. Similarly, dynamic correlations alone, though capturing important motion patterns, cannot fully characterize the binding pocket geometry, leaving room for additional improvements. The integration of both information types enables the model to simultaneously consider geometric constraints and dynamic interaction patterns, achieving better performance across both error metrics and correlation coefficients and demonstrating the value of this combined approach.

The consistent improvement pattern across multiple archi-

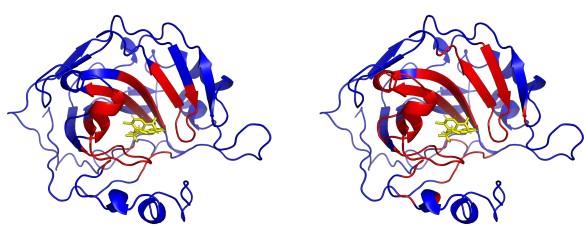

*Figure 3.* **Binding Site Prediction**. Visualization of binding site detection task using protein structure (PDB ID 3M67). Left: Ground truth binding site residues (shown in red) and the bound ligand (yellow). Right: Predicted binding site residues using the Combined Graph, showing reasonable agreement with the true binding regions for this example. This case illustrates how the model aims to identify residues within 10 Åfrom the ligand. The ligand is shown only for reference and is not provided to the model.

tectures validates that these performance gains arise from the complementary nature of static and dynamic features rather than specific architectural choices. These results show that the fusion of both static and dynamic information enhances the accuracy of binding affinity prediction, offering valuable insights for drug design and molecular screening applications.

### 4.3.4. EFFECT OF TRAJECTORY ALIGNMENT

To validate the impact of trajectory alignment on our correlation-based representations, we compare model performance using aligned versus unaligned trajectories on the atomic adaptability prediction task, as it most directly reflects the quality of our dynamic information capture. Table 4 shows that the aligned Correlation Graph consistently outperforms its unaligned counterpart across all metrics. Using RGCN, alignment improves Pearson correlation from 0.5493 to 0.6426 and reduces MAE from 0.2591 to 0.2311. The Aligned Combined Graph, which integrates aligned correlations with distance information, also shows substantial performance improvements, with RGCN achieving a Pearson correlation of 0.7326 (vs 0.6762 unaligned) and MAE of 0.1981 (vs 0.2192 unaligned). Similar comprehensive improvements are observed with RGAT for both Correlation and Combined graphs, where alignment consistently enhances performance across all metrics. These comprehensive improvements across both architectures and graph types demonstrate that removing global rigid-body motions effectively isolates meaningful conformational dynamics, leading to more accurate predictions.

## 5. Conclusion

This work addresses a key limitation in current protein graph representations: their exclusive reliance on static structural information without incorporating crucial information about protein dynamics. We propose a novel heterogeneous graph representation that integrates static structural information and dynamic correlations from molecular simulation trajectories, and apply relational graph neural networks to process these enriched representations. Our systematic evaluation across diverse architectures examines three distinct tasks: atomic adaptability prediction, binding site detection, and binding affinity prediction. The experimental results show that while Distance and Correlation graphs exhibit different performance patterns across architectures and tasks, the Combined Graph consistently achieves superior performance across all tasks, metrics, and architectures. These results demonstrate that static and dynamic information provide complementary signals for understanding protein behavior. Our approach opens new possibilities for protein modeling and design by effectively capturing both static structural constraints and dynamic correlations in a unified framework.

Future directions include exploring advanced architectures like graph transformers to enhance heterogeneous information processing, and investigating additional correlation measures such as mutual information to enrich dynamic feature representation. As a broader direction, integration with emerging generative models for molecular dynamics could further expand the applicability of our approach by trajectory generation, especially when molecular dynamics trajectories are not readily available. These developments will further strengthen our approach's capability in protein modeling, advancing applications in drug design and structural biology.

## Acknowledgements

We thank the reviewers for their constructive feedback that helped improve the quality of this work. This research was conducted at the Laboratory of Signal Processing 2 and the Laboratory of Protein Design & Immunoengineering at École Polytechnique Fédérale de Lausanne, and Wageningen University & Research. B.C. acknowledges support from the Swiss National Science Foundation.

## Impact Statement

This paper presents work whose goal is to advance the field of Machine Learning, specifically in protein modeling. Our approach for integrating static and dynamic protein information could contribute to drug discovery and protein engineering applications. There are many potential societal consequences of our work, none of which we feel must be specifically highlighted here.

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

# A. Appendix

## A.1. Trajectory Alignment

During preprocessing, we aligned all molecular dynamics trajectories to their initial frames using PyTraj's `align` function (Roe & Cheatham III, 2013). The alignment eliminates global translations and rotations, ensuring that $\Delta \mathbf{r}_i^t$ captures meaningful conformational changes rather than rigid-body motions. By focusing on intrinsic protein dynamics, this preprocessing step improves the quality of our correlation-based edges and leads to more informative graph representations.

## A.2. Implementation Details and Hyperparameters

We implemented our models using PyTorch Geometric. Each model consists of 5 GNN layers followed by a two-layer MLP for prediction. We trained models using the Adam optimizer with a learning rate of 1e-4 and batch size of 32. Training epochs were task-specific: 50 for atomic adaptability prediction, 200 for binding site detection, and 500 for binding affinity prediction.

For model architecture optimization, we explored different hidden dimensions for each model-task-graph type combination, with detailed results presented in Tables 5, 6 and 7, . The dimension ranges were selected based on architectural differences and memory constraints. For example, in atomic adaptability prediction using RGCN, we explored hidden dimensions {26, 32, 53, 64}, while for RGAT we tested {17, 20, 24} due to its higher memory requirements. For R-EGNN and R-GPS, we tested hidden dimensions {32, 64} across all tasks. For R-SS-GNN, we followed the original SS-GNN hyperparameters and set the hidden dimension to 108.

*Table 5.* **Detailed Results for Atomic Adaptability Prediction with Different Hidden Dimensions**. Supplementary results to Table 1, showing the performance of different hidden dimensions for each model and graph type combination. Values represent individual runs ($\downarrow$ indicates lower is better, $\uparrow$ indicates higher is better).

| MODEL | GRAPH TYPE | HIDDEN DIM | MAE ($\downarrow$) | RMSE ($\downarrow$) | PEARSON R ($\uparrow$) | SPEARMAN R ($\uparrow$) |
|---|---|---|---|---|---|---|
| RGCN | DISTANCE | 26 | 0.2636 | 0.4347 | 0.5008 | 0.5292 |
| | | 32 | 0.2726 | 0.4361 | 0.4960 | 0.5216 |
| | | 53 | 0.2670 | 0.4423 | 0.5134 | 0.5409 |
| | | 64 | 0.2687 | 0.4412 | 0.5160 | 0.5454 |
| | CORRELATION | 26 | 0.2359 | 0.3965 | 0.6130 | 0.6790 |
| | | 32 | 0.2397 | 0.3951 | 0.6170 | 0.6817 |
| | | 53 | 0.2316 | 0.3884 | 0.6336 | 0.6909 |
| | | 64 | 0.2321 | 0.3874 | 0.6359 | 0.6967 |
| | COMBINED | 21 | 0.2074 | 0.3530 | 0.7115 | 0.7753 |
| | | 26 | 0.2077 | 0.3548 | 0.7073 | 0.7732 |
| | | 32 | 0.2078 | 0.3549 | 0.7084 | 0.7778 |
| | | 44 | 0.2039 | 0.3481 | 0.7206 | 0.7824 |
| | | 53 | 0.2016 | 0.3475 | 0.7214 | 0.7854 |
| | | 64 | 0.2034 | 0.3433 | 0.7301 | 0.7910 |
| RGAT | DISTANCE | 20 | 0.2768 | 0.4406 | 0.4790 | 0.4811 |
| | | 24 | 0.2763 | 0.4426 | 0.4715 | 0.4750 |
| | CORRELATION | 20 | 0.2461 | 0.3976 | 0.6104 | 0.6496 |
| | | 24 | 0.2465 | 0.3984 | 0.6084 | 0.6504 |
| | COMBINED | 17 | 0.2085 | 0.3536 | 0.7098 | 0.7555 |
| | | 20 | 0.2067 | 0.3538 | 0.7096 | 0.7578 |
| | | 24 | 0.2161 | 0.3557 | 0.7088 | 0.7615 |

## A.3. Dataset Details

For atomic adaptability and binding site detection tasks, we used the data splitting in the MISATO dataset splits, with 13,597 samples for training, 1,582 for validation, and 1,593 for test. These splits were created using sequence-based clustering with BlastP (similarity threshold of 30%) to prevent information leakage through structural similarities. The dataset contains molecular dynamics trajectories generated using the Amber20 software package with a simulation length of 10ns.

*Table 6.* **Detailed Results for Binding Affinity Prediction with Different Hidden Dimensions**. Supplementary results to Table 3, showing the performance of different hidden dimensions for each model and graph type combination. Values represent individual runs (↓ indicates lower is better, ↑ indicates higher is better).

| MODEL | GRAPH TYPE | HIDDEN DIM | RMSE (↓) | MAE (↓) | PEARSON R (↑) | SPEARMAN R (↑) |
|---|---|---|---|---|---|---|
| RGCN | DISTANCE | 32 | 1.7036 | 1.3441 | 0.6444 | 0.6147 |
| | | 64 | 1.6377 | 1.2750 | 0.6740 | 0.6486 |
| | CORRELATION | 32 | 1.7497 | 1.4077 | 0.6204 | 0.5965 |
| | | 64 | 1.5761 | 1.2053 | 0.7025 | 0.6921 |
| | COMBINED | 25 | 1.6186 | 1.2465 | 0.6772 | 0.6666 |
| | | 32 | 1.6454 | 1.3366 | 0.6650 | 0.6570 |
| | | 51 | 1.6150 | 1.2999 | 0.6821 | 0.6678 |
| | | 64 | 1.5996 | 1.2243 | 0.6860 | 0.6718 |
| RGAT | DISTANCE | 32 | 1.7134 | 1.3655 | 0.6330 | 0.6141 |
| | | 64 | 1.6194 | 1.2626 | 0.6875 | 0.6478 |
| | CORRELATION | 32 | 1.6456 | 1.3295 | 0.6803 | 0.6651 |
| | | 64 | 1.6670 | 1.3527 | 0.6555 | 0.6150 |
| | COMBINED | 26 | 1.7224 | 1.3993 | 0.6237 | 0.5948 |
| | | 32 | 1.7016 | 1.3542 | 0.6389 | 0.6196 |
| | | 55 | 1.6411 | 1.3309 | 0.6704 | 0.6563 |
| | | 64 | 1.5685 | 1.2378 | 0.7075 | 0.6855 |

*Table 7.* **Detailed Results for Binding Site Detection with Different Hidden Dimensions**. Supplementary results to Table 2, showing the performance of different hidden dimensions for each model and graph type combination. Values represent mean ± standard deviation over 5 runs (↑ indicates higher is better).

| MODEL | GRAPH TYPE | HIDDEN DIM | ACC (↑) | PRECISION (↑) | RECALL (↑) | F1 SCORE (↑) |
|---|---|---|---|---|---|---|
| RGCN | DISTANCE | 32 | 0.7112 ± 0.0092 | 0.1678 ± 0.0024 | 0.4464 ± 0.0164 | 0.2428 ± 0.0027 |
| | | 64 | 0.7217 ± 0.0136 | 0.1694 ± 0.0031 | 0.4270 ± 0.0223 | 0.2412 ± 0.0018 |
| | CORRELATION | 32 | 0.7282 ± 0.0069 | 0.1808 ± 0.0022 | 0.4552 ± 0.0102 | 0.2578 ± 0.0012 |
| | | 64 | 0.7206 ± 0.0033 | 0.1784 ± 0.0007 | 0.4652 ± 0.0079 | 0.2569 ± 0.0014 |
| | COMBINED | 26 | 0.7433 ± 0.0067 | 0.2005 ± 0.0030 | 0.4889 ± 0.0111 | 0.2834 ± 0.0023 |
| | | 32 | 0.7527 ± 0.0054 | 0.2042 ± 0.0037 | 0.4748 ± 0.0125 | 0.2847 ± 0.0044 |
| | | 53 | 0.7649 ± 0.0100 | 0.2083 ± 0.0029 | 0.4477 ± 0.0244 | 0.2829 ± 0.0033 |
| | | 64 | 0.7640 ± 0.0063 | 0.2086 ± 0.0025 | 0.4531 ± 0.0133 | 0.2846 ± 0.0018 |
| RGAT | DISTANCE | 32 | 0.6447 ± 0.0232 | 0.1453 ± 0.0034 | 0.4607 ± 0.0327 | 0.2084 ± 0.0036 |
| | | 48 | 0.6602 ± 0.0120 | 0.1475 ± 0.0032 | 0.4439 ± 0.0234 | 0.2089 ± 0.0040 |
| | CORRELATION | 32 | 0.6938 ± 0.0111 | 0.1664 ± 0.0031 | 0.4441 ± 0.0182 | 0.2294 ± 0.0030 |
| | | 48 | 0.6955 ± 0.0122 | 0.1653 ± 0.0036 | 0.4379 ± 0.0169 | 0.2279 ± 0.0031 |
| | COMBINED | 27 | 0.7226 ± 0.0067 | 0.1861 ± 0.0029 | 0.4750 ± 0.0137 | 0.2574 ± 0.0032 |
| | | 32 | 0.7291 ± 0.0148 | 0.1882 ± 0.0072 | 0.4637 ± 0.0218 | 0.2564 ± 0.0075 |
| | | 40 | 0.7367 ± 0.0055 | 0.1916 ± 0.0018 | 0.4563 ± 0.0133 | 0.2594 ± 0.0037 |
| | | 48 | 0.7387 ± 0.0103 | 0.1884 ± 0.0077 | 0.4360 ± 0.0033 | 0.2516 ± 0.0071 |

For binding affinity prediction, following previous work (Li et al., 2021a), we used the PDBbind 2020 refined set for training and validation, and evaluated on the core set. The refined set consists of 5,316 protein-ligand complexes specifically selected for high-quality binding data and crystal structures through a comprehensive filtering process (Liu et al., 2017). This dataset construction ensures reliable binding affinity values derived from carefully curated experimental measurements.

### A.4. Atomic adaptability

Atomic adaptability ($\gamma_x$) for each atom $x$ is calculated as the mean distance from its initial position across all simulation frames after alignment:

$$\gamma_x = \frac{1}{N_{\text{frames}}} \sum_{i}^{N_{\text{frames}}} \|\mathbf{r}_{\text{ref},x} - \mathbf{r}_{i,x}\| \tag{9}$$

where $\mathbf{r}_{\text{ref},x}$ is the initial position of atom $x$ and $\mathbf{r}_{i,x}$ is its position in frame $i$. This measure quantifies each atom's mobility throughout the simulation, providing insight into conformational flexibility at atomic resolution.

While atomic adaptability can be directly computed from molecular dynamics trajectories, predicting it from graph representations provides a valuable benchmark for evaluating how effectively different graph structures capture dynamic information. The Correlation Graph represents a compressed encoding of the full trajectory information, so the ability to accurately predict adaptability demonstrates that this encoding successfully preserves essential dynamic features. This task also maintains continuity with the established MISATO benchmark, facilitating direct comparison with current and future approaches.

### A.5. Architecture Performance Analysis

We analyzed the performance of invariant (RGCN, RGAT) and equivariant (R-EGNN) graph neural networks across our three tasks and graph types. R-GPS and R-SS-GNN are excluded from this comparison as they represent different paradigms (graph transformer and domain-specific model, respectively). Table 8 summarizes the best-performing architecture for each scenario.

*Table 8.* Best-performing GNN for each task and graph type combination

| Task | Distance Graph | Correlation Graph | Combined Graph |
|---|---|---|---|
| Atomic Adaptability Prediction | R-EGNN | R-EGNN | RGCN |
| Binding Site Detection | R-EGNN | R-EGNN | R-EGNN |
| Binding Affinity Prediction | R-EGNN | RGAT | RGCN |

Several patterns emerge from this analysis. R-EGNN generally outperforms invariant models across most scenarios, which is expected as equivariant architectures preserve rotational and translational symmetries crucial for protein modeling. R-EGNN consistently performs best on Distance Graphs across all tasks, which aligns with the fact that EGNN explicitly uses coordinates and distances in its message passing process, making them well-suited for distance-based representations. Binding site detection shows the most consistent benefit from equivariant architectures across all graph types, likely due to the more regular graph structure (all nodes represent $C\alpha$ atoms, although belonging to different amino acids), making 3D spatial relationships particularly important.

Interestingly, for Combined Graphs, invariant models (RGCN) outperform R-EGNN in two tasks. This suggests that our current implementation of relational EGNN may not optimally integrate information from different edge types, indicating opportunities for improved fusion mechanisms for equivariant architectures in future work.

### A.6. Theoretical Analysis

To understand why our Combined Graph approach consistently improves performance, we conducted additional theoretical analyses.

### A.6.1. GRAPH PROPERTIES

We analyzed graph properties of Distance and Combined graphs using randomly selected subsets of 50 proteins each at atomic and residue levels. Table 9 shows that adding correlation edges significantly reduces both graph diameter and average shortest path length at both levels, suggesting that correlation edges create critical shortcuts between dynamically coupled regions.

*Table 9.* Graph properties of Distance and Combined graphs

| Graph Level | Metric | Distance | Combined |
|---|---|---|---|
| Atomic | Diameter | 24.4 | 21.3 |
| Atomic | Avg. Shortest Path | 9.7 | 8.9 |
| Residue | Diameter | 10.1 | 6.7 |
| Residue | Avg. Shortest Path | 4.3 | 3.2 |

### A.6.2. GRAPH CURVATURE ANALYSIS

We analyzed Ollivier-Ricci curvature of Distance and Combined graphs using an example protein (PDB-ID 2I5J). Figure 4 shows that Combined graphs exhibit more positively curved edges, indicating reduced over-squashing and improved long-range information propagation in graph neural networks (Topping et al., 2021).

### A.7. Code Availability

Implementation will be made available at https://github.com/PKGuo/ protein-static-dynamic-fusion.git.

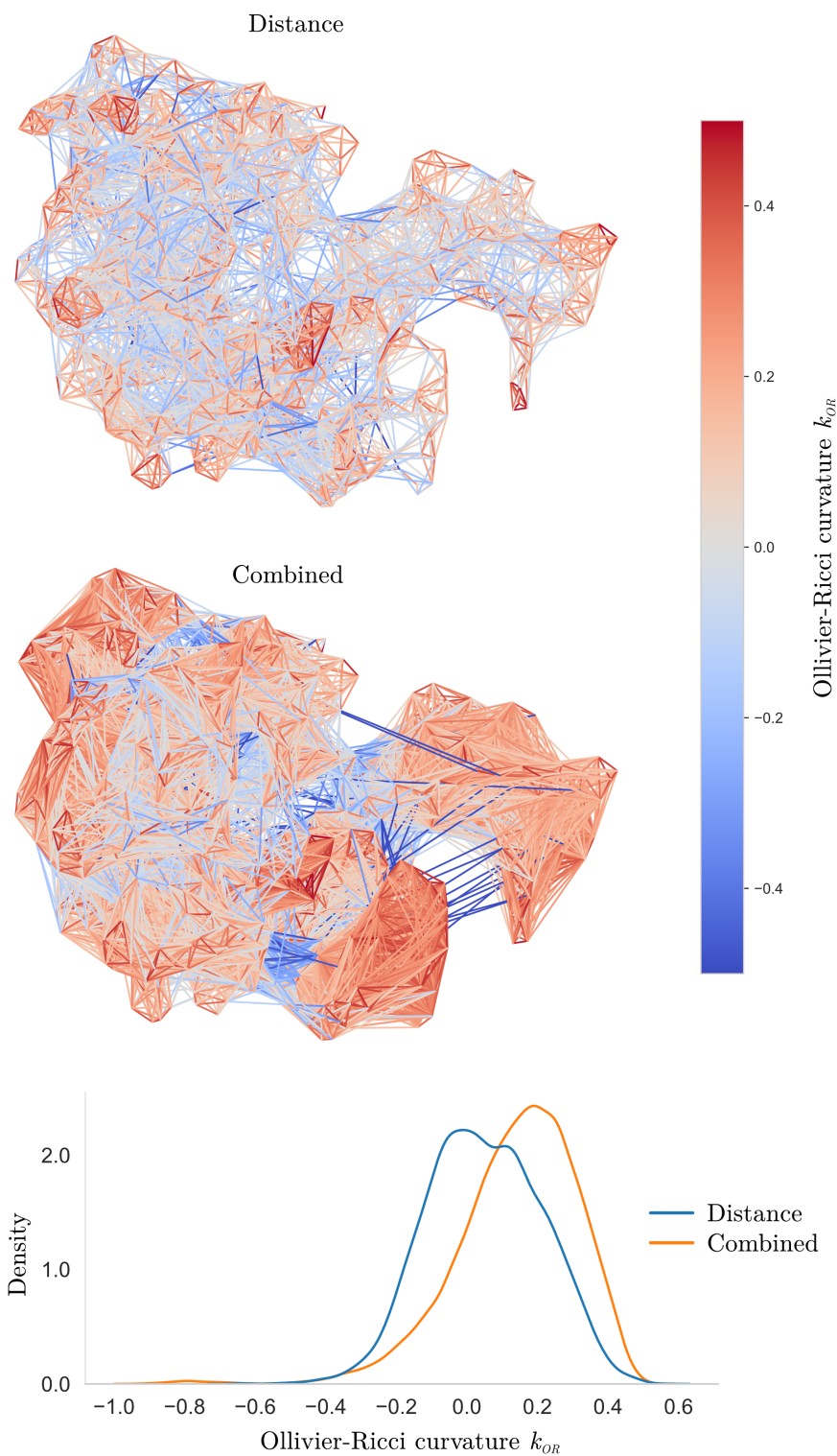

*Figure 4.* Ollivier-Ricci curvature analysis comparing Distance and Combined graphs for protein PDB-ID 2I5J. The color scale represents curvature values, with red indicating positive curvature and blue indicating negative curvature. The density plot shows the distribution of curvature values, showing that Combined graphs have more positively curved edges.

