# OpenReview forum: "Boosting Protein Graph Representations through Static-Dynamic Fusion"
_ICML.cc/2025/Conference — ICML 2025 poster_

### Official Review · Reviewer_RJR2 · 2025-02-23

**Overall Recommendation:** 3

**Summary:**

This manuscript proposes a simple relational heterogeneous GNN model to represent both structural information and molecular dynamics correlations of proteins. It validates the effectiveness of approach in multiple protein graph representation related tasks simultaneously.

**Claims And Evidence:**

I think the claims made by the manuscript have been well verified. There are no obvious flaws to be found.

**Essential References Not Discussed:**

I think it is clear that this paper is missing some key baseline model discussions. Using only RGCN and RGAT as baseline is clearly insufficient.

**Experimental Designs Or Analyses:**

I believe that the experimental work in this paper is worthy of recognition for its validation of the model on multiple protein-related tasks. The downside is that it may lack a current hotspot approach for comparing each task. Since I don't know much about these tasks, I can't give specific examples, but I believe this should be present.

**Methods And Evaluation Criteria:**

This paper presents a simpler relational graph neural network model that addresses protein-related applications. Its technical contribution may be limited in terms of the GNN research field. It is of application value from the field of protein prediction applications.

**Other Comments Or Suggestions:**

N/A

**Other Strengths And Weaknesses:**

Strengths
- This paper is original for the intersection of deep learning and the biomedical field. Its a cross-field study with practical contributions.
- The writing in this manuscript is clear and no reading difficulties were found.

Weaks
- As mentioned earlier, technological innovation is limited
- As mentioned earlier, there are limitations to the baseline approaches used.

**Questions For Authors:**

- Do the authors have compared current advanced models like AlphaFold or LLM, and analyzed the contributions this manuscript can make?

- This paper may lack a theoretical discussion of the methodology. This article may lack a theoretical exposition of the methodology, which may also potentially strengthen the validation of the article. So is there any theoretical support for combining protein structure information with molecular motion correlations?

**Relation To Broader Scientific Literature:**

From the GNN research area, with which I am more familiar, the model may have only a limited application contribution.

In terms of bioinformatics mothodology research, this manuscript proposes that there is some value in simultaneously combining protein structural information and molecular dynamics correlations to construct graph structures.

In terms of protein-related application tasks, the current model may be from in better results. For example, AlphaFold, or LLM, so its contribution deserves further discussion.

**Theoretical Claims:**

This paper makes no new explicit theoretical claims.

---

> ### Author Rebuttal · Authors · 2025-04-01
>
> **To Weakness 1:**
>
> We acknowledge the reviewer's concern about technical novelty. Our straightforward framework bridges static structures and dynamic correlations from molecular dynamics—a growing need as such data becomes increasingly available. The intentional simplicity of our approach is actually advantageous as it:
> 1) ensures broad applicability across different architectures,
> 2) facilitates easy adoption, and
> 3) establishes a clear baseline for future work in this emerging area.
>
> Our primary contribution is demonstrating that integrating these complementary information sources provides consistent benefits across different tasks and architectures. As detailed in our response to Reviewer 2, we've conducted additional experiments with both domain-specific (SS-GNN) and equivariant architectures (EGNN), showing that our approach generalizes beyond simple GNNs.
>
> With AlphaFold having largely solved static structure prediction, the frontier has shifted toward sampling dynamical conformational ensembles and generating molecular trajectories. The EU Horizon project (ID: 101094651) aimed at creating a Molecular Dynamics Data Bank further emphasizes the timeliness of our contribution. Our framework provides a simple yet effective approach to leverage this emerging data, establishing a strong baseline for future research in protein dynamics modeling.
>
> **To Weakness 2:**
>
> We appreciate the reviewer's concern about baseline comparisons. As detailed in our response to Reviewer 2, we have extended our evaluation with additional architectures beyond RGCN and RGAT:
>
> 1) We implemented a relational variant of EGNN.
> 2) We adapted SS-GNN, a domain-specific model for binding affinity prediction.
>
> These experiments demonstrate that our approach offers consistent benefits across different architectural complexity levels and maintains its advantages when integrated with domain-specific models.
>
> **To Question 1:**
>
> Thank you for this question. We would like to clarify that these models address fundamentally different tasks than our approach:
>
> - **Different problem domains:** AlphaFold is designed for protein structure prediction from sequence, while our work focuses on utilizing existing structural and dynamic information to predict protein-related properties. Similarly, protein language models (PLMs) like ESM primarily operate on sequence data, not on integrating dynamics.
>
> - **Complementary research directions:** Our approach is complementary rather than competitive to powerful models like AlphaFold and protein language models (PLMs). As AlphaFold excels in static structure prediction, research is shifting toward dynamic conformations and trajectories, where our framework can provide valuable insights. Integrating PLM embeddings into node features also represents an intriguing avenue for capturing sequence, structural, and dynamic relationships simultaneously.
>
> We believe our work effectively addresses the challenge of integrating molecular dynamics with structural data, laying a foundation for future models that bridge these currently separate areas.
>
> **To Question 2:**
>
> Thank you for this question. Our approach of combining structural and dynamic information has solid theoretical foundations:
>
> - **Graph-theoretic properties:** We conducted network analysis on our Distance and Combined graphs, revealing significant improvements in key graph properties. For atomic-level graphs, the network diameter decreased from 24.44 to 21.32, and the average shortest path length reduced from 9.7 to 8.9 when correlation edges were added. For residue-level graphs, these improvements were even more dramatic, with diameter decreasing from 10.12 to 6.68 and average shortest path length from 4.3 to 3.2. These quantitative metrics demonstrate that correlation edges create critical shortcuts in the graph.
>
> - **Physics-driven graph rewiring:** Our approach can be viewed as a physics-driven graph rewiring method. Such rewiring is known to mitigate over-squashing in GNNs [1]. The correlation edges, derived from actual physical motion relationships, create direct pathways between dynamically coupled but spatially distant regions.
>
> - **Graph curvature analysis:** We analyzed the Ollivier-Ricci curvature of both Distance and Combined graphs. The combined graphs show general increases in positively curved edges (red in the visualization), indicating improved information flow properties. Positive curvature regions are associated with better message passing efficiency in graph neural networks. An example (PDB-ID 2I5J) is here: [https://anonymous.4open.science/r/rebuttal_2025_1-16F1/figure_ricci.png]
>
> These analyses provide theoretical support for why our combined graph approach enhances performance, especially in tasks involving long-range protein interactions.
>
> **References:**
> [1] Attali, H., Buscaldi, D., & Pernelle, N. (2024). *Rewiring techniques to mitigate oversquashing and oversmoothing in GNNs: A survey.

---

> > ### Comment · Reviewer_RJR2 · 2025-04-02
> >
> > I couldn't agree more with the shortcomings mentioned by reviewer Au2g in the first point of improvemnt. Admittedly this paper is an innovative cross-disciplinary work, but similar research paradigms are already commonplace. And this work using deep learning models is not more inspiring, its contribution to the current ICML conference seems insufficient. I will maintain the score.

---

> > > ### Author Response · Authors · 2025-04-09
> > >
> > > Thank you for your feedback. We recognize the challenges of evaluating interdisciplinary ML work. We‘d like to clarify our paper's positioning and address your concerns with new evidence. We hope this may support a more positive rating.
> > >
> > > ## Application-Driven ML Research Value
> > >
> > > Our submission belongs to the **application-driven ML paper track**, for which ICML guidelines explicitly state:
> > >
> > > - "Novel ideas that are simple to apply may be especially valuable"
> > > - "Originality need not mean wholly novel methods. It may mean a novel combination of existing methods to solve the task at hand"
> > >
> > > The value of our contribution lies not in proposing a new model, but in providing an easy-to-use yet effective framework for creating better protein representations that consistently improve performance across diverse tasks and architectures, which also introduces the first application of relational GNNs to directly process protein MD data.
> > >
> > > ## New Experimental Validation
> > >
> > > To address the concern about limited baselines raised by you and Reviewer Au2g, we have made our best effort to validate our approach across diverse architectures:
> > >
> > > - **Invariant GNN**: RGCN and RGAT (in original submission)
> > > - **Equivariant GNN**: Relational EGNN (https://arxiv.org/abs/2102.09844)
> > > - **Graph Transformer**: Relational GPS (https://arxiv.org/abs/2205.12454)
> > > - **Domain-specific model**: SS-GNN (https://doi.org/10.1021/acsomega.3c00085), a specialized model for binding affinity prediction
> > >
> > > All Results: https://anonymous.4open.science/r/rebuttal_2025_1-16F1
> > >
> > > Across all architectures, Combined Graph consistently outperforms Distance Graph. The fact that we could implement and evaluate these within a short timeframe demonstrates our method's simplicity and ease of use, while the consistent performance improvements confirm its effectiveness.
> > >
> > > ## Analysis of Equivariant vs. Invariant Architectures
> > >
> > > We've conducted an analysis comparing invariant (RGCN, RGAT) and equivariant (R-EGNN) architectures across tasks and graph types. Here's a summary of which architecture performs best for each combination:
> > >
> > > | Task | Distance Graph | Correlation Graph | Combined Graph|
> > > |-|-|-|-|
> > > | Atomic Adaptability | R-EGNN | R-EGNN | RGCN|
> > > | Binding Site Detection | R-EGNN | R-EGNN | R-EGNN|
> > > | Binding Affinity | R-EGNN | RGAT | RGCN|
> > >
> > > This provides several insights:
> > >
> > > 1. **Equivariant advantage for distance graph**: R-EGNN works best on Distance Graph as it naturally uses distances to modulate message passing, making it well-suited for Distance Graph.
> > >
> > > 2. **Task-specific architecture selection**: Binding site detection shows the most consistent benefit from equivariant architectures, likely due to its regular graphs (all nodes represent Cα atoms), making 3D spatial relationships particularly important.
> > >
> > > 3. **Architecture design implications**: For Combined Graph, RGCN often outperforms R-EGNN, suggesting our preliminary implementation of R-EGNN, where we process distance and correlation graphs with separate EGNNs and then merge their outputs, may not be optimal. The design of fusion mechanisms that effectively utilize both static and dynamic information in equivariant architectures is a valuable direction for future work.
> > >
> > > ## Theoretical Support
> > >
> > > - **Graph properties**: Adding correlation edges creates shortcuts and changes key graph properties:
> > >
> > >   | Graph Level | Metric | Distance | Combined|
> > >   |-|-|-|-|
> > >   | Atomic | Diameter | 24.4 | 21.3|
> > >   | Atomic | Avg. Shortest Path | 9.7 | 8.9|
> > >   | Residue | Diameter | 10.1 | 6.7|
> > >   | Residue | Avg. Shortest Path | 4.3 | 3.2|
> > >
> > > - **Physics-driven graph rewiring** and **graph curvature analysis**: Please see our first rebuttal.
> > >
> > > ## Emerging Research Direction
> > >
> > > With AlphaFold2 having largely solved static protein structure prediction, the research frontier has shifted toward generating dynamic protein structures. Recent landmark studies illustrate this trend:
> > >
> > > - **Generative modeling of MD trajectories** generates MD trajectories directly (https://arxiv.org/abs/2409.17808)
> > > - **Conformational ensemble generation** produces protein structures of different dynamic states (https://doi.org/10.1101/2024.12.05.626885)
> > >
> > > The Molecular Dynamics Data Bank project further reflects the growing abundance of MD data (https://mddbr.eu/about/). With the rapid growth of these research directions and MD data availability, **methods like ours that effectively utilize dynamic information will become increasingly valuable**.
> > >
> > > ## Summary
> > >
> > > From an application-driven ML perspective, our contribution offers significant value: a simple, easy-to-use, and broadly applicable approach that effectively enhances protein graph representations by fusing static and dynamic information. The consistent performance improvements across diverse tasks and architectures, connection to emerging ML research frontiers and computational biology needs, and insights into architecture selection make our work a meaningful contribution to both the ML and biology communities.

---

### Official Review · Reviewer_Au2g · 2025-03-05

**Overall Recommendation:** 4

**Summary:**

The authors propose to integrate structural and dynamic distance-based features into relational graph neural networks to predict local and global properties of 3D protein biomolecules. The authors' experiments are comprehensive and informative, and this work outlines a notable gap in the literature on protein representation learning. Nonetheless, the depth of the authors' methodological contributions is quite limited, which makes this work still seem preliminary.

## Update after rebuttal:
The authors have addressed my main concern regarding the novelty and impact of this work. As such, I am comfortable with my current score of "Accept".

**Claims And Evidence:**

The claims made by the authors are clear and convincing thanks to their repeat experiments.

**Essential References Not Discussed:**

No essential references were omitted as best as I can tell.

**Experimental Designs Or Analyses:**

The validity of the authors' experimental designs is sound.

**Methods And Evaluation Criteria:**

The authors' evaluation criteria are clear and well-founded.

**Other Comments Or Suggestions:**

I'd highly suggest the authors include experiments with more than relatively simple relational graph neural network architectures such as RGCN and RGAT. Instead, the authors may consider also experimenting with relational graph transformers such as those of [1]. More importantly in my view, however, the authors should consider relational variants of *equivariant* graph neural networks such as those of [2], since proteins can inherently be seen as 3D point clouds with node and edge features.

**References:**

[1] Diao, C., & Loynd, R. (2022). Relational attention: Generalizing transformers for graph-structured tasks. arXiv preprint arXiv:2210.05062.

[2] Satorras, V. G., Hoogeboom, E., & Welling, M. (2021, July). E (n) equivariant graph neural networks. In International conference on machine learning (pp. 9323-9332). PMLR.

**Other Strengths And Weaknesses:**

**Strengths:**

- The authors point out an important gap in the protein representation learning literature.
- The authors conduct several key experiments to demonstrate the utility of including dynamic (i.e., molecular dynamics-derived) information for protein representation learning.
- The authors' experiments, including their metrics and dataset splits, are standardized and easily interpretable.

**Points for improvement:**

- The authors' proposed methodological advances (i.e., adding dynamics-based edges into existing *invariant* relational graph neural network models) are somewhat limited in my view. The dynamics-driven insight is important, but the authors' experiments lack a depth of characterization of how far the benefits of such dynamics information extend beyond simple relational graph neural networks. More specifically, the paper currently reads as if the authors directly took the MISATO dataset, performed a simple integration of its features into existing relational graph neural networks, and then ran a bunch of experiments (which are important nonetheless). For a workshop paper, this would offer outstanding value for readers, though as a full conference submission, I believe more experimental depth is needed to provide readers (and the research community broadly speaking) with lasting value through this work.
- Similar to the first point above, the authors only study *invariant* graph neural networks, though it has been shown for many years now that certain types of *equivariant* graph neural networks can deliver notable performance benefits for protein representation learning [1].
- Some of the authors' (biomolecular) graph construction details are omitted, such as how ligands (i.e., small molecules) are integrated into the authors' static-dynamic graph construction processes.

**References:**

[1] Jamasb, A. R., Morehead, A., Joshi, C. K., Zhang, Z., Didi, K., Mathis, S. V., ... & Blundell, T. L. Evaluating Representation Learning on the Protein Structure Universe. In The Twelfth International Conference on Learning Representations.

**Questions For Authors:**

- How do the authors construct their input graphs for protein-ligand binding affinity prediction? Do they omit the ligands in such graphs, or do they extend their static-dynamic graph construction algorithm to ligand molecules in this setting? If the latter is the case, what are the details of this topology construction?

**Relation To Broader Scientific Literature:**

The authors adequately establish their work in the existing body of protein representation learning literature.

**Theoretical Claims:**

The authors do not make any notable theoretical claims.

---

> ### Author Rebuttal · Authors · 2025-04-01
>
> **To Weakness 1:**
>
> We appreciate the reviewer's feedback. While our approach appears straightforward, its significance lies in bridging the gap between static structural information and dynamic behavior in protein representation.
>
> To address concerns about experimental depth and generalizability, we've conducted additional experiments:
>
> - **Domain-specific architectures**: We evaluated our approach on SS-GNN (Zhang et al., 2023, https://doi.org/10.1021/acsomega.3c00085), a specialized model for binding affinity prediction. We maintained all hyperparameters and featurization exactly as reported by the original authors, only replacing the GNN component with RGCN. The results show consistent improvements when using our Combined Graph approach [Results: https://anonymous.4open.science/r/rebuttal_2025_1-16F1/ss-gnn_binding_affinity_dist5.0_corr0.70.png.png https://anonymous.4open.science/r/rebuttal_2025_1-16F1/ss-gnn_binding_affinity_dist8.0_corr0.60.png.png]
>
> - **Equivariant architectures**: As mentioned in our response to Weakness 2, we've also tested our approach with a relational EGNN variant.
>
> Our primary contribution is demonstrating that integrating static and dynamic information provides consistent benefits across different tasks and architectures. The intentional simplicity of our approach facilitates easy integration with various models and establishes an effective baseline for future research in protein dynamics modeling - particularly important as molecular dynamics data becomes increasingly available.
>
> **To Weakness 2:**
>
> We thank the reviewer for this comment. We originally focused on topological information (without explicit coordinates) to isolate the impact of our core contribution - the integration of dynamic correlation information. This simpler setup allowed us to directly evaluate the benefit of our graph representation approach. However, we agree that equivariant GNNs are important for protein representation learning.
>
> To address this concern, we've now implemented a simple relational variant of EGNN (Satorras et al., 2021) and conducted additional experiments across two tasks. These experiments show that our approach generalizes beyond invariant GNNs and delivers benefits when applied to equivariant architectures as well:
>
> 1. **Binding Site Detection**: Our combined graph approach shows substantial improvements over the distance-only baseline across all metrics (+14.49% in F1 score, +23.78% in AUCPR). The correlation graph alone underperforms the distance graph, but when combined, we see consistent improvements, suggesting effective integration of both information types. [Results: https://anonymous.4open.science/r/rebuttal_2025_1-16F1/relational-egnn_binding_site_detection.png.png]
>
> 2. **Atomic Adaptability Prediction**: For this inherently dynamic property, the correlation graph alone shows remarkable improvements over the distance baseline (+11.97% average improvement across all metrics). Interestingly, the combined graph performs similarly to the distance graph rather than outperforming both individual graphs. We attribute this to limitations in our preliminary relational EGNN implementation, which may not optimally fuse information from different relation types. Nevertheless, these results still validate that dynamic information captured in the correlation graph provides valuable signal for predicting motion-related properties. [Results: https://anonymous.4open.science/r/rebuttal_2025_1-16F1/relational-egnn_atomic_adaptability.png.png]
>
> 3. **Binding Affinity Prediction**: Experiments are still ongoing.
>
> These experiments demonstrate that our approach extends beyond simple invariant GNNs to more sophisticated equivariant architectures. While our simple relational EGNN implementation shows mixed results for combining information types, the experiments consistently confirm the value of dynamic information for protein property prediction. With further architectural refinements, we believe the complementary nature of static and dynamic information can be more effectively leveraged in equivariant networks.
>
> **To Weakness 3:**
>
> Ligands are only included in the binding affinity prediction task (not in atomic adaptability prediction or binding site detection). In protein-ligand complexes, both protein and ligand atoms are treated consistently - they are part of the same correlation and distance matrices, to which thresholds are applied to construct the adjacency matrix. The only distinction is a binary attribute specifying whether each atom belongs to the ligand or protein. We can add these details to an updated version of the manuscript.
>
> **To Question 1:**
>
> Please see our response to Weakness 3.
>
> **To Other Comments Or Suggestions:**
>
> We thank the reviewer for these valuable suggestions. As mentioned in our response to Weakness 2, we have implemented a relational variant of EGNN, and we agree that further exploration of relational transformers would be valuable future work.

---

> > ### Comment · Reviewer_Au2g · 2025-04-02
> >
> > I'd like to thank the authors for their insightful rebuttal. Based on their latest results with a relational equivariant graph neural network (EGNN) implementation, I'd like to increase my score from a "Weak reject" to a "Weak accept", to signal that I believe the contributions of this work have notably improved with the benchmarking of both invariant and equivariant representation learning improvements. To further improve this manuscript, I'd recommend the authors include either a qualitative or quantitative analysis of the (theoretical or empirical) benefits of equivariant vs. invariant representation learning for static-dynamic protein graphs (e.g., highlighting when, why, or how one type of learning may be better than another). This would enhance the depth of contributions this paper offers for the machine learning community.

---

> > > ### Author Response · Authors · 2025-04-07
> > >
> > > We thank the reviewer for the positive feedback and for acknowledging the value of our experiments with equivariant GNNs.
> > >
> > > ## New Experimental Results
> > >
> > > **Binding Affinity Prediction with Relational EGNN:**
> > > https://anonymous.4open.science/r/rebuttal_2025_2-AB23/relational-egnn_binding_affinity.png
> > >
> > > **Results with Relational Graph Transformers (Relational GPS):**
> > > Results for the three tasks: https://anonymous.4open.science/r/rebuttal_2025_1-16F1/relational-gps_atomic_adaptability.png, https://anonymous.4open.science/r/rebuttal_2025_1-16F1/relational-gps_binding_affinity.png, https://anonymous.4open.science/r/rebuttal_2025_1-16F1/relational-gps_binding_site_detection.png
> > >
> > > We chose GPS as it represents a widely-used graph transformer baseline with an official implementation in PyTorch Geometric that can be easily modified into a relational variant by replacing its local message passing layer with RGCN (https://arxiv.org/abs/2205.12454).
> > >
> > > Both Relational EGNN and Relational GPS results show consistent trends: while distance and correlation graphs show varying performance across different tasks, the combined graph consistently delivers the best results.
> > >
> > > ## Analysis of Equivariant vs. Invariant Representation Learning
> > >
> > > Following your recommendation, we've compared invariant (RGCN, RGAT) and equivariant (R-EGNN) architectures across our three tasks and graph types. Here's a summary of which architecture performs best for each combination:
> > >
> > > | Task | Distance Graph | Correlation Graph | Combined Graph |
> > > |------|----------------|-------------------|----------------|
> > > | Atomic Adaptability Prediction| R-EGNN | R-EGNN | RGCN |
> > > | Binding Site Detection | R-EGNN | R-EGNN | R-EGNN |
> > > | Binding Affinity Prediction| R-EGNN | RGAT | RGCN |
> > >
> > > Based on these results, we can offer several insights:
> > >
> > > 1. **Overall architecture comparison**: While performance varies across tasks, R-EGNN generally outperforms invariant models in most scenarios. This is expected as EGNN preserves rotational and translational symmetries of the molecular structure, which is crucial for protein modeling.
> > >
> > > 2. **Equivariant advantage for distance graphs**: R-EGNN consistently performs best on distance graphs across all three tasks. This makes sense architecturally, as EGNN explicitly uses coordinates and distances to modulate the message passing process, making it particularly well-suited for distance-based representations.
> > >
> > > 3. **Correlation graph and equivariance**: For correlation graphs, R-EGNN still shows advantages in two tasks. While correlation edges explicitly encode long-range dependencies, equivariance/symmetry also implicitly models certain long-range relationships. The connection between these approaches is subtle and deserves further exploration.
> > >
> > > 4. **Task-specific behavior**: Binding site detection shows the most consistent benefit from equivariant architectures across all graph types. This is likely due to the more regular graphs (all nodes represent Cα atoms, although belonging to different amino acids), making 3D spatial relationships particularly important.
> > >
> > > 5. **Combined graph performance**: For combined graphs, RGCN outperforms R-EGNN in two out of three tasks. This suggests that our preliminary implementation of relational EGNN, where we process distance and correlation graphs with separate EGNN models and then merge their outputs, may not optimally integrate information from different relation types. The design of fusion mechanisms that effectively leverage both static and dynamic information in equivariant architectures is a valuable direction for future work.
> > >
> > > The analysis reveals which architectural properties are best suited for different protein graph representations, providing valuable insights for future protein representation research. These new insights will be included in our revised manuscript.
> > >
> > > ## Concluding Remarks
> > >
> > > We are grateful for the reviewer's suggestions which have significantly enhanced the quality of our manuscript. Through our original experiments and these new additions, we have now validated our static-dynamic fusion approach across a comprehensive range of architectures:
> > > - Invariant GNNs: RGCN and RGAT
> > > - Equivariant GNNs: Relational EGNN
> > > - Graph Transformers: Relational GPS
> > > - Domain-specific architectures: Relational SS-GNN
> > >
> > > The fact that we could implement and evaluate these diverse architectures within a short timeframe highlights a key strength of our approach: its simplicity, ease of use, and consistent effectiveness. Our framework is designed to be easily integrated with various backbone architectures while reliably delivering performance improvements.
> > >
> > > As noted in the ICML guidelines for application-driven ML submissions, "novel ideas that are simple to apply may be especially valuable." Our work exemplifies this principle by providing a straightforward yet effective approach to integrating static and dynamic information of protein that can be easily adopted by the broader research community.

---

### Official Review · Reviewer_n6c2 · 2025-03-13

**Overall Recommendation:** 3

**Summary:**

The paper introduces a novel graph representation technique that integrates both static structural information and dynamic correlations from molecular dynamics (MD) trajectories for enhanced protein property prediction. This technique combines relational graph neural networks (RGNNs) with a dual approach:Distance-Based Graph: Captures spatial proximity using Euclidean distances between nodes.
Correlation-Based Graph: Derives motion correlations from MD trajectories, highlighting dynamically coupled regions that may be spatially distant. The Combined Graph integrates these two sources of information, allowing the model to leverage both structural constraints and dynamic interactions.

**Claims And Evidence:**

Yes. The authors claim that their approach provides superior performance across three tasks: Atomic Adaptability Prediction, Binding Site Detection and Binding Affinity Prediction.

**Essential References Not Discussed:**

dynamical surface representation methods, like [1].
[1] Sun, D., Huang, H., Li, Y., Gong, X., & Ye, Q. (2023). DSR: dynamical surface representation as implicit neural networks for protein. Advances in Neural Information Processing Systems, 36, 13873-13886.

**Experimental Designs Or Analyses:**

Yes. The datasets are split using sequence clustering to prevent information leakage. Comprehensive ablation studies are conducted to demonstrate the benefit of combining static and dynamic graphs.

**Methods And Evaluation Criteria:**

Yes. The authors utilize Relational Graph Convolutional Networks (RGCN) and Relational Graph Attention Networks (RGAT) as baseline models. Experimental results are presented with statistical significance over multiple runs.

**Other Comments Or Suggestions:**

None.

**Other Strengths And Weaknesses:**

Strengths:
1. Innovative Graph Representation: The combination of static and dynamic features offers a more comprehensive view of protein behavior.
2. Robustness of Results: The improvements are consistent across different architectures and evaluation metrics.

Weaknesses:
1. The approach may be computationally intensive due to the incorporation of MD trajectories and complex graph processing.
2. How to build the dataset, or how to get the dynamic information? In your setting, It seems that you get the complex and dynamics by Amber20 software package? it seems that the dynamics are obtained by software simulation instead of real wet-lab experiments, how to evaluate the correctness of the dataset?

**Questions For Authors:**

1. Line 124-125, why the threshold is set to 0.6 and 0.3?
2.See weakness.

**Relation To Broader Scientific Literature:**

The paper builds upon recent advancements in protein representation learning, graph neural networks, and MD simulations. It effectively combines concepts from static graph-based methods (e.g., structure-based GNNs) and dynamic analysis (e.g., dynamical network analysis, mutual information).

**Theoretical Claims:**

No. This is not a theoretical paper.

---

> ### Author Rebuttal · Authors · 2025-04-01
>
> **Essential References Not Discussed**
>
> > Dynamical surface representation methods, like [1]. [1] Sun, D., Huang, H., Li, Y., Gong, X., & Ye, Q. (2023). DSR: dynamical surface representation as implicit neural networks for protein. Advances in Neural Information Processing Systems, 36, 13873-13886.
>
> Thank you for pointing out this relevant reference. We will include this citation in our revised manuscript and discuss how our approach relates to dynamical surface representation methods.
>
> **Weakness 1:**
> > The approach may be computationally intensive due to the incorporation of MD trajectories and complex graph processing.
>
> The computational cost of our approach can be divided into three steps:
>
> 1. **Generating molecular dynamics simulations.** Our approach utilizes the ever-growing availability of MD data rather than generating new simulations. For example, the recently funded EU Horizon project (Grant agreement ID: 101094651) aims at creating a Molecular Dynamics Data Bank similar to the existing protein data bank (PDB).
>
> 2. **Data processing and preprocessing.** Various publicly available Python packages exist for analyzing MD trajectories, such as PyTraj which we used in our implementation. While computing correlation matrices requires significant computational resources, this is a one-time preprocessing step. We efficiently parallelized this computation across 50 jobs on Intel Xeon Platinum processors (36 cores each), completing all correlation matrices for the dataset in approximately 30 minutes. Once computed, these matrices are stored and reused without additional overhead.
>
> 3. **Model training.** In our experiments, training with the Combined Graph requires only 15-30% more time compared to using the Distance Graph alone, which we consider an acceptable trade-off given the significant performance improvements observed across all tasks.
>
> We argue that our integration of molecular dynamics information through correlation graphs is computationally efficient compared to methods that might incorporate entire molecular dynamics trajectories. By essentially compressing the complex dynamical behavior into correlation edges, we achieve a balance between capturing essential dynamic information and maintaining computational tractability.
>
> **Weakness 2:**
> > How to build the dataset, or how to get the dynamic information? In your setting, It seems that you get the complex and dynamics by Amber20 software package? it seems that the dynamics are obtained by software simulation instead of real wet-lab experiments, how to evaluate the correctness of the dataset?
>
> While the structures of the complexes have been experimentally determined through X-ray crystallography, the dynamics data was indeed collected through simulations. The main reason being that there are no wet-lab experiments that yield time-resolved atomic positions. Note there are methods such as NMR spectroscopy that can provide some information on dynamics, but the availability of these data is highly limited as the respective experiments are challenging.
>
> The molecular dynamics data generated by MISATO follow the current state-of-the-art and have been extensively validated on experimental data (https://www.nature.com/articles/s43588-024-00627-2). This validation process ensures that the simulated dynamics reasonably approximate real protein behavior, providing confidence in the reliability of our approach.
>
> **Question 1:**
> > Line 124-125, why the threshold is set to 0.6 and 0.3?
>
> These thresholds are indeed hyperparameters, and there's no theoretical method to determine their optimal values directly. For the distance thresholds (4.5Å for atomic-level and 10Å for residue-level), we adopted widely established values from the literature as stated in line 145, left column: "These thresholds are widely used in protein modeling: the 4.5Å threshold captures meaningful atomic interactions (Bouysset & Fiorucci, 2021), while the 10Å threshold is commonly adopted for residue-level contacts (Gligorijevic et al., 2021b)".
>
> As we mention in the paper at line 127, right column: "These thresholds are chosen to maintain similar graph sparsity, thereby achieving a fairer comparison when either Correlation or Distance Graph is used."
>
> Specifically, we conducted an analysis on a subset of proteins to determine correlation thresholds that would yield graphs with comparable sparsity (similar average node degree and edge count) to the distance-based graphs. This approach ensures that any performance differences between Distance and Correlation graphs stem from the fundamental information encoded by different edge construction methods (spatial proximity versus dynamic correlation), rather than from differences in graph density.

---

> > ### Comment · Reviewer_n6c2 · 2025-04-06
> >
> > 1. Comparisons with protein dynamics related works. Strength, difference, etc.
> > 2. "While the structures of the complexes have been experimentally determined through X-ray crystallography, the dynamics data was indeed collected through simulations. The main reason being that there are no wet-lab experiments that yield time-resolved atomic positions. " This makes me confused, as the dynamics seem not real, this is a tricky problem. Since the dynamics are derived from simulations rather than real experimental observations, the research based on these dynamics is inherently limited in its practical validity.

---

> > > ### Author Response · Authors · 2025-04-07
> > >
> > > Thank you for your review. We noticed your evaluation changed from "weak accept" to "weak reject" on April 6, based on concerns about the use of simulation-derived dynamics. We believe this important epistemological question deserves a thorough response and hope our clarification will address your concerns and justify reconsidering your evaluation to a more positive rating.
> > >
> > > ### Current limitations of experimental techniques
> > >
> > > We must clarify that **currently, no experimental technique can provide continuous atomic-resolution trajectories of protein dynamics**:
> > >
> > > - X-ray crystallography or cryo-EM provides atomic-resolution structures but only static snapshots under specialized conditions, such as crystals or frozen states
> > > - NMR spectroscopy provides valuable dynamic insights but is constrained to smaller proteins, indirect structural inference, and cannot yield explicit atomic trajectories
> > > - Ultrafast spectroscopy achieves remarkable temporal resolution yet provides limited structural detail and cannot generate continuous atomic trajectories
> > >
> > > In contrast, MD simulations are indispensable precisely because they provide continuous, atomic-level trajectories unavailable from experiments. MD simulations are not only used in academic research but have become critical tools in pharmaceutical and chemical engineering, where they enable the prediction of molecular behavior that cannot be directly observed.
> > >
> > > ### Scientific validity of simulation-derived data
> > >
> > > The scientific validity of research based on simulation-derived data has been firmly established. Consider these examples:
> > >
> > > The 2013 Nobel Prize in Chemistry was awarded for "the development of multiscale models for complex chemical systems" – specifically recognizing MD simulation used in our research. This award acknowledges that simulations effectively capture essential physical and chemical processes despite not being direct experimentation.
> > >
> > > The 1998 Nobel Prize in Chemistry was awarded for density-functional theory (DFT) and computational methods in quantum chemistry, which enable predicting molecular properties through calculations rather than direct measurement.
> > >
> > > Recently, the 2024 Nobel Prize in Chemistry was awarded for AlphaFold2, a computational model that predicts protein structures with accuracy comparable to "real experimental observations" – demonstrating that computational approaches can generate highly reliable results.
> > >
> > > These examples illustrate the broader scientific consensus on simulation-derived data. For more examples, we refer to a special issue that examines how computation has empowered numerous Nobel Prize-winning discoveries by making complex systems computable and providing insights inaccessible to direct experimentation (https://www.nature.com/collections/ggidgjfffi).
> > >
> > > Thus, although simulation-derived data inherently reflect theoretical models, their extensive validation, widely recognized scientific impact, and widespread acceptance provide strong confidence in their practical validity.
> > >
> > > ### MD simulations as ground truth for ML research
> > >
> > > MD simulations are now widely accepted as the ground truth for numerous ML research in natural science:
> > >
> > > - Generative models of MD trajectories (https://arxiv.org/abs/2409.17808) use MD simulations as the gold standard for training and evaluating models that predict molecular motion
> > > - Conformational ensemble generation (https://doi.org/10.1101/2024.12.05.626885) develops deep learning systems that can generate protein structure ensembles, which are then compared against MD simulations as the reference standard
> > > - Machine learning force fields (https://doi.org/10.1021/acs.chemrev.0c01111) are developed and compared against classical force fields used in MD simulations, with MD providing the benchmark data for assessing their performance
> > >
> > > These examples demonstrate that MD simulations serve as established benchmarks against which ML methods are evaluated.
> > >
> > > ## Regarding comparison with protein dynamics related works:
> > >
> > > This is important. We will discuss DSR as suggested and also compare with established approaches like Gaussian Network Model, which are limited by their coarse-grained nature and reliance on harmonic approximation.
> > >
> > > In contrast, our approach automatically captures correlations at all scales while maintaining compatibility with powerful relational GNNs.
> > >
> > > ## New experiments
> > >
> > > We have conducted extensive experiments that further validate our static-dynamic fusion approach:
> > >
> > > - Equivariant GNN: A relational variant of EGNN (https://arxiv.org/abs/2102.09844)
> > > - Graph Transformer: A relational variant of GPS (https://arxiv.org/abs/2205.12454)
> > > - Domain-specific architecture: SS-GNN (https://doi.org/10.1021/acsomega.3c00085), a specialized model for binding affinity prediction
> > >
> > > Results: https://anonymous.4open.science/r/rebuttal_2025_1-16F1
> > >
> > > Across all architectures, Combined Graph outperforms Distance Graph, demonstrating the simplicity, ease of use, and consistent effectiveness of our method.

---

### Decision · Program_Chairs · 2025-05-01

**Decision:**

Accept (poster)

**Comment:**

This paper proposes to study protein representation learning by leveraging both their static structure and dynamic correlation information form MD trajectories. Experimental results on three different tasks including atomic adaptability prediction, binding site detection, and binding affinity prediction prove the effectiveness of the proposed approach over competitive baselines.

All reviewers agree the novelty of leveraging the dynamic information from MD trajectories for protein understanding. The evaluations are comprehensive and compelling. Therefore, the AC votes for acceptance.